

# The measurement and impact of light absorbing particles on snow surfaces

Carl G. Schmitt[1,2,3], Bria L. Riggs[4], Ulyana N. Horodyskyj[5], Alia L. Khan[1], Holly A. Ewing[4], John D. All[1,3], Wilmer Sanchez Rodriguez[3]

[1]Environmental Science Dept. Mountain Environments Research Institute, Western Washington University, Bellingham, WA, USA
[2]Mesoscale and Microscale Meteorology Division, National Center for Atmospheric Research, Boulder, CO, USA
[3]American Climber Science Program, Nederland, CO, USA
[4]Bates College, Lewiston, ME, USA
[5]Science in the Wild, Broomfield, CO, USA

*Correspondence to*: Carl G Schmitt (schmitt.carlg@gmail.com)

**Abstract.** Light absorbing particles (LAPs) can have a significant impact on the albedo of snow. LAPs absorb solar radiation which warms surrounding snow thereby increasing melt or sublimation rates. Historically, LAP concentrations have been reported in terms of a mass mixing ratio, typically in nanograms of black carbon per gram of snow. While this

representation is convenient for sampling, it can lead to deceptive results if there is significant surface accumulation of LAPs due to snow loss or dry deposition. Here we demonstrate that LAPs concentrated on the snow surface can substantially affect the albedo and typical sampling strategies and reporting protocols can lead to highly erroneous estimates of albedo.

Theoretical calculations and measurements both show that the reduction in albedo by LAPs can be twice as strong when particles are concentrated on the surface as opposed to being mixed within the top thin layer of snow. Current commonly used

sampling strategies are not sufficient to determine the necessary information to assess the impact of surface LAPs on snowpack albedo.

To facilitate more accurate albedo estimates, we propose a new sampling strategy to better characterize LAP distribution in and on snowpacks. Theoretical calculations and experimental measurements show that snowpack albedo can be much better characterized when using the suggested sampling strategy to determine the distribution of LAPs present.

**1 Introduction**

Light absorbing particles on snow have classically been measured and reported in terms of a mass mixing ratio (MMR) of LAPs, for example as nanograms of black carbon (or effective black carbon, eBC, Grenfell et al., 2011) per gram of snow. Sampling strategies can include the collection of snow at specific depths in a snow pit (Doherty et al., 2013), or sections of an ice core (Ginot et al., 2014), or simply surface and sub-surface measurements where the depth of the surface snow sample is

loosely defined (Schmitt et al., 2015). The results of these studies are difficult to compare due to the lack of uniformity in snow density and sampling depth.





Studies have reported MMRs of LAPs on the order of more than 1000 ng g$^{-1}$ eBC (Xu et al., 2012; Ming et al., 2016), although it is much more common for MMRs to be less than 100 ng g$^{-1}$ for fresh snow. As snow melts or sublimates, LAPs generally do not go away and can collect on the surface of the snowpack, which in extreme cases can be obvious if the surface

snow layer is removed (Fig. 1). As shown in Ming et al., (2016), the albedo of the snow can decrease dramatically through the dry season as LAPs accumulate on the surface or as dirty surfaces are exposed.

LAPs can become part of the snowpack through three basic processes: 1) Precipitation processes can bring LAPs to the snowpack through scavenged by falling precipitation or LAPs can serve as condensation or ice nuclei, both of which result in LAPs mixed within the snowpack; 2) LAPs can be deposited by dry deposition, in which case LAPs accumulate on the surface

of the snow; 3) LAPs can be concentrated at the surface when LAP-laden snow melts or sublimates, leaving the LAPs behind, typically on the surface.

While situations similar to those depicted in Fig. 1 are uncommon, they can easily arise in certain situations. Glaciers in regions with extended dry seasons experience snow loss throughout the dry season through melt or sublimation and can easily accumulate LAPs in a two-dimensional layer on the surface. This applies to seasonal snow as well when most of the winter of

LAPs can accumulate on the surface as snow melts away in the Spring. Regions with substantial dry deposition, for example the dust events commonly reported in southern Colorado, USA (Painter et al., 2010, Skiles et al., 2012) or substantial LAP deposition due to forest fires (Delaney et al., 2015), can also result in abundant LAPs in a two-dimensional surface layer.

The commonly used online version of the Snow, Ice, and Aerosol Radiation (SNICAR) model (Flanner et al., 2007, Flanner et al., 2009) uses MMR for radiative transfer calculations of the impact of LAPs in snow. For the aforementioned very

high MMR cases, it is likely that the bulk of the LAPs are on the surface of the snow and not uniformly distributed throughout the snowpack. Optically, surface versus sub-surface LAPs should be treated separately, as surface LAPs reduce or stop light from entering the snowpack and the absorbed energy is at the ice-air interface where it can efficiently melt or sublimate snow.

Here, we demonstrate that it is extremely important to understand the LAP concentrations "in" the snow and "on" the snow in order to effectively estimate albedo from LAP measurements. In section 2, theoretical calculations of radiative

processes in a snowpack are used to illustrate that both surface-layer LAP concentration and snow-borne LAP concentration can strongly affect albedo calculations. Section 3 presents results of spectral albedo measurements where the LAPs were initially concentrated in a surface layer, then mixed into the snow. In section 4 we present a new sampling strategy that enables the determination of the surface LAP concentration as well as the sub-surface MMR leading to a more complete understanding of LAP distribution, thus improving albedo estimates and our understanding of melt dynamics. The results are summarized

and sampling strategy recommendations for different conditions are given in the final section.

## 2 Theoretical Justification

In many instances, an MMR is reasonable measure of LAP concentration. However, in situations where surface layers are common due to extended dry seasons or significant dry deposition, such as in the Cordillera Blanca in Peru (Fig. 1), a bulk



snow estimate of MMR derived from, for example the top 5 cm of snow, explains only part of the story, and can lead to

significant error in calculated albedo estimates. In unpublished data from Vallunaraju mountain in the Cordillera Blanca, Peru region in 2016, one measurement showed an MMR eBC value of 8000 ng g$^{-1}$ for a surface measurement (sampled by collecting the top 1 cm of snow) while the snow sample collected directly below (1-5 cm) had MMR eBC values near 200 ng g$^{-1}$. Had the "surface" sample been collected to a depth of 2 cm rather than 1, the MMR for the sample could have been about half what was sampled (assuming uniform density) showing the arbitrariness of reporting an MMR when the LAPs are concentrated in

a surface layer.

Traditional measurement techniques provide a three-dimensional (volume based) result when at times a two-dimensional (area based) result is necessary for most accurate assessment of albedo. This can be understood by considering the processes of energy transfer in a snowpack with LAPs both on the surface and in the sub-surface (Fig. 2, processes A-H). When sunlight reaches the air-snow interface (A), a portion is immediately reflected without interacting with the snow (B). An additional

component is absorbed by the surface LAPs (C). Of the light that passes through the two-dimensional surface and enters the snow (D), the portion that is not absorbed in the snow (E) is re-directed back out of the snow (F) and subjected to additional loss as it crosses the air snow interface if there is a surface layer (G). The final albedo is the ratio of the sum of the amount radiated after interaction with the snowpack (H) and that reflected immediately (B) divided by the incident radiation (A). Thus, a large surface accumulation of LAPs will greatly enhance processes C and G, and result in a much smaller value at H and,

hence, a much lower albedo.

To define the extent of the apparent difference in albedo for cases where a substantial surface LAP layer is present versus the LAPs being well mixed, we calculate the anticipated albedo for several different real and hypothetical situations (Table 1). Two scenarios are presented based on the previously described sample from Vallunaraju mountain in the Cordillera Blanca, Peru (Vallu 1 and 2 in Table 1) which had a very heavy surface layer and comparatively few LAPs in the snow below (Fig. 1).

Three scenarios are shown for the "Next to Mine 7" site in Svalbard described in Khan et al., 2017 (Sval 1, 2, and 3 in Table 1) where there was not only a heavy surface layer of LAPs but also high concentrations of LAPs within the snow. In Svalbard case, snow samples of the top 10 cm of snow were collected as well as spectral albedo measurements. As the collected snow was a mixture including the surface layer, we identified three possible LAP distributions that would lead to the measured MMR and used those scenarios to estimate range of albedo values that could be implied. Utilizing these field measurements and

possible scenarios based on them, we calculated albedo using four methods: using SNICAR-Online and the measurement MMR, using SNICAR-Online with the sub-surface LAP measurement, with full treatment of the surface and sub-surface, and the same but assuming a lower Mass Absorption Cross section (MAC) for the LAPs.

To rigorously estimate the albedo for snow with a surface layer of LAPs, the following calculations are done. For demonstration purposes, we step through the calculations for the Vallu 1 sample (values shown in column 1 of Table 1). The

sample collected showed an MMR of 8000 ng g$^{-1}$ eBC for a 1 cm deep sample, and by estimating that the surface snow density was 0.5 g cm$^{-3}$, we estimate that the surface layer was approximately 0.04 grams of eBC per square meter at interface A-C in Fig. 2. The snow sample was collected and analyzed using the Light Absorbing Heating Method (LAHM, Schmitt et al., 2015,



Schmitt et al., 2019) and thus the eBC value is referenced to Fullerene soot type BC. Using a MAC of 9 $m^2$ $g^{-1}$ of Fullerene soot BC (Linke et al., 2016), this suggests that the light blocking capacity of the surface layer of eBC is equal to 0.34 $m^2$ $m^{-2}$

of snow thus 34% of the incoming light is absorbed by the surface layer of LAPs. Note that if the LAPs were perfectly spread, it would be 0.36 $m^2$ $m^{-2}$, but we adjust this to 0.34 because random distribution of particles would lead to some overlapping particles. Before the incoming sunlight interacts with the snow, the surface LAPs already reduced the possible light reflection to 66% at interface A-C (Fig. 2, process C). Of the 66% that interacts with snow rather than the surface LAPs, a portion is reflected immediately by the surface of the snow (process B), and a portion enters the snow. As the asymmetry parameter (the

ratio of the light scattered into the forward hemisphere by a particle versus the total light scattered) of hexagonal crystals and spheres is approximately 0.8 to 0.85, it can be estimated that 15 to 20% of the energy that interacts with non-LAP covered snow is directly reflected up by the surface. For demonstration purposes, we will say a factor of 0.15 of the 66% of remaining energy (10% of the original incoming total) that interacts with the snow is reflected from the surface and contributes to the overall albedo, leaving 56% of the initial light to enter the snowpack.

In the snow (Fig. 2, location D), the 56% of the initial light that enters the snowpack is either absorbed within the snowpack (Fig. 2, process E) or redirected back to the surface interface (Fig. 2, process F). As the sub-surface snow was found to have an MMR of ~200 ng $g^{-1}$ eBC we can use the SNICAR model to estimate the absorption within the snowpack. The SNICAR-Online model predicts that the broadband albedo would be approximately 0.80 for an MMR of 200 (assuming large crystals and very deep snow/ice). As we are only interested in what is being absorbed in the snow, we need to account for the

energy that SNICAR-Online would assume was reflected up without interaction (which we estimated in the previous paragraph with the asymmetry parameter), thus increasing the energy in the snow that can be reflected back to the surface. Of the initial 100%, 56% makes it into the snow, 23% of that is absorbed within the snow (Fig. 2, process E), thus 43% (a 23% reduction of the 56%) returns to the surface (Fig. 2 process F). The 23% reduction can be verified by doing the above calculations but assuming that there is no surface layer. The impact of the surface reflection is subtracted, then added back in resulting in the

SNICAR-Online albedo being restored to the value calculated for the sub-surface snow.

The 43% of energy that returns to the surface of the snow is again subjected to the 34% reduction due to the surface layer (Fig. 2 process G) leading to 29% of the initial energy having entered the snow will exit the snow. This, combined with the 10% that was reflected at the surface (process B), leads to 39% of the incident energy being reflected or an albedo of 0.39. In contrast, by using the SNICAR-Online calculation with an MMR of 8000 ng $g^{-1}$ eBC, the broadband albedo estimate was

0.635, a reduction from clean snow values of less than half of that expected when the surface layer is considered. This is highly dependent on sampling strategy (Table 1, Vallu 2), as a 4 cm deep sample would led to an MMR of ~2200 ng $g^{-1}$ eBC which, using SNICAR-Online alone, would have led to an estimate of 0.74 for the broadband albedo. Thus, it would be possible through common sampling strategy variation to calculate that the albedo could be between 0.74 and 0.63 when consideration of the surface layer alone leads one to conclude that the albedo would be closer to 0.39 with about half of the incoming energy

having been absorbed at the surface.





Mathematically, the above discussion can be reduced so that the albedo of snow with a significant surface layer of LAPs can be calculated using Equation 1:

$$\alpha = \sigma \times [R_s + \sigma \times (\alpha_s - R_s)] \tag{1}$$

where $\alpha$ is albedo, $\alpha_s$ is the albedo of snow without the surface layer of LAPs; $\sigma$ is the fractional area that is not covered by LAPs in units of m$^2$ m$^{-2}$; and $R_s$ is the surface reflectance of clean snow.

Thus, if $A$ is area covered by LAPs, $\sigma = 1 - A$ where $A$ is a measure of the absorption cross section of the LAPs and can be estimated by multiplying the LAP concentration in g m$^{-2}$ of the surface layer by the MAC. $R_s$ is the fraction of light that is immediately reflected at the surface without penetrating into the snowpack. In the previous discussion this was estimated as 1.0 minus the approximate asymmetry parameter of a sphere of similar size to the snow grain size (0.15). The albedo of the snow, assuming that there is not a surface layer, $\alpha_s$, can be estimated using SNICAR-Online and the sub-surface LAP MMR.

Note that if $A$ is substantial (i.e. greater than 0.1 m$^2$ m$^{-2}$) then $A$ should be reduced due to the possibility of LAPs overlapping other LAPs. By simulating a surface being randomly covered with small particles with overlapping being possible, it was found that the total area covered could be approximated by:

$$A_{corr} = \frac{A}{\sqrt{1+A^2}} \tag{2}$$

where $A_{corr}$ is the actual area covered by particles with the total area $A$, the reduction being caused by the random chance of LAP particles overlapping when randomly distributed.

Columns 1 and 2 of Table 1 show the results of the Vallu calculations including "simple" albedo calculations from SNICAR-Online and the measured surface MMR (to 1 cm depth) as well as estimates of albedo using the MMR estimated from sampling the top 4 cm. The remainder of Table 1 shows three calculations from a study by Khan et al., (2017) based on LAP measurements from Svalbard, Norway, near an active coal mine. For this set of examples, the spectral albedo was also measured and showed a substantially reduced albedo (Hemispherical Directional Reflectance Factor to a value of 0.16). Snow samples were collected for the top 10 cm of snow with no attempt to separate surface LAPs from sub-surface LAPs. The LAHM technique showed that LAPs MMR was 5000 ng g$^{-1}$ in the top 10 cm of snow. With the MMR determined for the 10 cm sample, we determined three different LAP distributions (surface layer and sub-surface layer pairs) scenarios for the top 10 cm of snow that would result in the 5000 ng g$^{-1}$ MMR for a fully mixed sample. The three Sval columns in Table 1 show these three different LAP estimates that would all lead to 5000 ng g$^{-1}$ MMR but have distinctly different broadband albedo values. Our calculations show that the scenario where the surface layer was estimated to have 0.08 g m$^{-2}$ and the sub-surface value was estimated to have 3000 ng g$^{-1}$ eBC (Sval 2) led to the measured LAP concentration as well as resulted in an albedo estimate using Equation 1 similar to the measured value of around 0.16. Values of surface eBC as low as 0.05 g m$^{-2}$ (with the corresponding higher sub-surface values; Sval 1) led to broadband albedo estimates of 0.31 close to twice those measured, while increasing the surface eBC to 0.12 g m$^{-2}$ (Sval 3) led to substantially lower albedo estimates (0.086). Note that the final row in Table 1 presents the results using a significantly reduced estimate of the MAC which could be a factor if LAPs clumped together as part of the melt freeze cycle.



The results in Table 1 demonstrate two important points. First, if there is a surface layer of LAPs of any significance, it
needs to be accounted for in order to calculate a reasonable albedo from the LAP measurements. This can be seen by comparing
the results of "simple albedo" to the albedo calculated using Equation 1 (rows 5 and 7). Second, it is important to know the
actual surface layer value since estimating the surface layer coverage from a three dimensional MMR value can still lead to
substantial uncertainty even when the sub-surface value is highly polluted as well.

Each of the necessary parameters in Equation 1 has an associated uncertainty range. For investigating uncertainty, we
assume that an uncertainty of 0.005 in calculated albedo is noteworthy.  In order to change the albedo by 0.005, a surface layer
needs to be approximately 350 micrograms eBC per square meter. Using the sampling strategy described below in section 4,
this quantity should be easily detectable with any common instrumentation, thus this uncertainty is likely to have little effect
on calculated albedo. $R_s$ is the reflectance of the top layer of snow crystals on the surface that are not covered by LAPs. With
a surface layer absorption equivalent to 0.1 $m^2$ $m^{-2}$, an asymmetry parameter change of 0.05 (0.85 to 0.80) leads to a change
of 0.005 in the calculated albedo, again, a not very substantial uncertainty. In Equation 1, $\alpha_s$, the SNICAR-Online albedo for
sub-surface snow, could be a substantial source of uncertainty, but its uncertainty is highly dependent on the measured MMR
(with lower MMRs having lower uncertainties) as well as additional uncertainties associated with the snow properties required
as inputs to SNICAR-Online. For determination of albedo differences due to LAPs (ie. keeping all other parameters the same
in the calculation except for LAPs), the uncertainties are substantially less than not considering the surface layer.

## 3 Experimental investigation

Given the magnitude of apparent uncertainty in calculations of albedo resulting from incomplete knowledge of the surface
LAP concentration, we were compelled to attempt to experimentally validate these results. To do this, we conducted a set of
experiments where we measured the spectral albedo of snow before and after spreading LAPs on a snow surface, then again
after mixing LAPs into the snow. The experiments were performed at 3000 m on Pikes Peak in Colorado, USA, in an area
with full sun and a snowpack depth of approximately 1 m. The air temperature was approximately -5°C and the wind was
moderate. Spectral albedo measurements were taken with a Malvern Panalytical (formerly, ASD) FieldSpec HandHeld 2
spectroradiometer. Each experimental area was prepared by first removing a wind-packed crust from an area approximately
1-meter square. The resulting surface was lightly mixed and smoothed. A white reference plate was placed in the experimental
area and a calibration spectrum was taken with the FieldSpec. That measurement was followed by an albedo measurement of
the reference plate, and then of the snow in the middle of the experimental area. For experimental LAPs, Solaray brand
activated coconut charcoal powder was used. Particles were spread using a common handheld small particle dispersion device
(EverydayLiving brand item # 60941 salt shaker). An attempt was made to cover a large portion of the experimental area with
LAPs, but the wind hampered these efforts. Typically, it was possible to identify an area 20-30 cm in diameter with a coating
uniform to the eye. The FieldSpec was centered about 30 cm above this area for the "LAPs on surface" measurement. The
LAPs in the uniform area were then mixed into the snowpack to a depth of approximately 3 cm using a homemade rake, and

this area was smoothed again to the same texture as the first measurement. The FieldSpec was then used to take a final measurement. The sun angle was such that the shadow of the spectrometer was always well away from the area being measured. While the described procedure is unlikely to lead to perfect closure between LAP distribution and albedo, the measurements were meant to qualitatively corroborate the calculations in the previous section.

Four experiments were conducted and an effort was made to achieve a variety of different LAP levels. Snow samples were collected after the final spectrometer reading was taken in order to assess the quantity of LAPs present. The LAHM technique was used for analysis of the samples (Schmitt et al., 2015 and 2019). The total LAP concentration was assumed to be 100% on the surface for the "LAPs on surface" measurement, then uniformly mixed into the top 3 cm of snow for the "LAPs mixed" measurement. A snow sample collected before spreading any LAPs showed that the background LAP

concentration (11.7 ng g$^{-1}$ eBC) in the snow was negligible compared to the experimental LAPs added (Table 2, rows 4 and 5). Figure 3 shows the spectral albedo measurements. From top to bottom in all of the plots the measured spectral albedo lines are always in the same order. The top (blue) line shows the spectral albedo of the white reference plate, the second (orange) line shows the spectral albedo of the snow before application of LAPs, the third (red) line shows the spectral albedo with the LAPs mixed into the snow, and the fourth and lowest (green) line shows the spectral albedo when the LAPs were on the surface

of the snow.

The spectral albedo measurements uniformly showed reductions in the albedo of the snow after the addition of LAPs, with the spectral albedo reduction being greatest for the measurements when the LAPs were on the surface compared to after the LAPs were mixed in (Fig. 3, Table 2). All four measurements showed that the spectral albedo of the snow was reduced by up to a factor of two more when the LAPs were on the surface as compared to when they were mixed into the snow. Visible

integrated albedo estimates (Table 2, rows 2 and 3), made by integrating the spectral albedo measurements multiplied by a 5777K Planck curve across the 450-950 nm range, were substantially lower than those from the SNICAR-Online broadband albedo estimated from the measured MMR values (Table 2, row 6). Uniformly, the albedo increased substantially after the particles were mixed with the mixed measurements recovering an average of 40% of the lost albedo. Although conditions likely hampered highly accurate results, the experiments clearly strengthen the hypothesis that a clear understanding of the

distribution of LAPs at or near the snow surface is critical for understanding LAP effects on albedo; albedo reductions from LAPs on the snowpack were much larger than albedo reductions when the LAPs were mixed into the snow.

## 4 Proposed measurement strategy

The results of the previous sections lead us to conclude that a new sampling strategy is necessary to better assess the distribution of LAPs on the surface of snow. Unless it is actively snowing, it is possible that a surface layer of LAPs has

developed on snow surfaces which can have a substantial impact on albedo. The technique described here assumes a filtering technique is being used for measurement of LAPs such as the LAHM technique described in Schmitt et al., (2015). It can be adapted to other instrumentation by adjusting the area sampled and volume of snow collected.



The new measurement strategy requires two samples be collected at each location to assess the surface layer LAP concentration and sub-surface MMR. The purpose of the first measurement is to determine the quantity of LAPs on the surface layer per unit area to calculate $\sigma$ in Equation 1. For the first measurement, use a ruler or other item of fixed length to measure an area for sampling (30 x 30 cm works well for the LAHM method). This size is just an example and could be expanded or reduced if the snow were particularly clean or dirty or based on the instrumentation to be used. Carefully scrape surface snow from the measured area and collect it into a suitable sample container. Collect all of the snow that could be subject to surface deposition of LAPs (a few millimeters at most for smooth surfaces, but more if the snow surface is rough). The purpose of the second sample is to determine the MMR of the sub-surface snow, necessary for determining $\alpha_s$ in Equation 1. For the second sample, collect snow from the same area, but below the surface being careful to not contaminate the sample with surface snow.

When processing the samples, the goal is to determine $A$ from equation 2 and the MMR of sub-surface snow to use in SNICAR-Online for the determination of $\alpha_s$. The necessary result from the first sample is the total quantity of LAPs in the sample, then that quantity will be related to the area sampled. This can be done by determining the MMR in the surface sample, then multiplying it by the total mass of the snow sample resulting in the total mass for the area measured. The total LAP concentration determined from the first sample is an over-estimate of the LAP concentration on the surface because some sub-surface snow was collected in the processes of collecting the surface LAPs. By definition, any snow collected in sample 1 is sub-surface snow as the surface layer of LAPs is defined as being on top of any snow. To correct for this, the MMR of the sub-surface snow can be used (the second sample). Analyze the second sample to determine the MMR of the sub-surface snow. Finally, to correct the surface total LAP concentration, assume that any snow collected with the surface sample had an MMR equal to that of the sub-surface snow. Subtract this residual by subtracting the snow mass collected in the first sample multiplied by the MMR determined for the second sample. This gives a total effective mass of LAPs in the surface layer only. The assumption is that the LAPs on the surface are of negligible mass when compared to the snow collected, and therefore all of the mass collected with the surface sample has the same MMR as the sub-surface snow. The LAP concentration on the surface can then be determined by dividing the total surface LAP concentration by the measured area when collecting sample 1. This surface LAP concentration, paired with the SNICAR-Online calculated sub-surface albedo enables the determination of the overall albedo using equation 1. A document (in multiple languages) detailing this sampling strategy along with instructional videos will be posted on our website: http://www.naturalsystemsresearch.com/calculate.html.

## 5 Conclusions

LAPs can occur in high concentration on the surface of snow through mechanisms ranging from dry deposition to surface accumulation as a result of snow loss by melting or sublimation. On the surface, LAPs can have a much stronger impact on the albedo as compared to LAPs mixed within the snowpack. Typical sampling strategies do not separate the surface LAPs from LAPs mixed in the snow, thus reducing their utility for albedo calculations. The measurement strategy described herein



provides a means for quantifying the surface and sub-surface LAP concentrations separately, information that can facilitate
more accurate estimation of albedo.

Recommendations: The processes that lead to a snowpack having a surface layer of LAPs all involve time. Thus, if
substantial time has passed since the previous snowfall, it could be important to sample in a manner so as to be able to quantify
LAPs on the surface layer. Freshly fallen snow generally will not have a surface layer unless the precipitation event was
followed by rapid dry deposition such as a dust storm, or possibly if the precipitation event ended with a period of rain. These
factors should all be considered when sampling, and if a surface layer is possible, the sampling strategy described in section 4
should be considered. More complicated snowpacks can easily evolve where a layer of LAPs formed during a prolonged period
of dry weather is subsequently covered by a thin layer of fresh snow. In this case, an MMR sample can be collected of the
newly fallen snow, then this snow could be brushed off of the old surface which can be measured as described in section 4
thus providing information on the snowpack before the newly fallen snow or after the fresh snow has melted or been blown
off. (This 3-sample method is suggested as it is very common in the mountains near to where the lead author lives.) Much
more complicated snowpacks (with numerous layers near the surface) could be sampled similarly but further model
developments would be necessary to utilize the additional information provided, likely with diminishing returns with deeper
sampling.

*Code availability*: Python code for the analysis of samples collected with the sampling strategy described in section 4 is
available upon request from CGS.

*Data availability*: Although this is not a data heavy publication, data for the plots shown in Fig. 3 are available from CGS upon
request.

*Author contributions:* CGS led the development of the theoretical study. BR assisted with the theoretical development and was
partially responsible for figure production. UH assisted by CGS conducted the spectral albedo portion of the study. ALK
assisted with sample studies leading to the development of the results and provided the Svalbard sample data. HE advised BR
and provided valuable guidance on the project and editing of the manuscript. JA and WSR led sample collection that ultimately
as well as conducted experiments which led to the development of the ideas presented. All authors assisted in writing and
editing the manuscript.

*Acknowledgements:* The authors would like to thank the contributors to and volunteers for the American Climber Science
Program (ACSP). Measurements collected during ACSP expeditions were the motivation for this publication. Thanks to
Malvern Panalytical (formerly ASD) for providing use of the HandHeld 2 spectroradiometer. Funding from the Bates College
Summer Research Fellowship and the Maximillian E. and Marion O. Hoffman Foundation assisted BR in conducting this





research. Additionally, the American Alpine Club and The Northface providing funding through the "Live Your Dream" grant, to BR which helped support field data collection.

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




Table 1

| | | Vallu 1 | Vallu 2 | Sval 1 | Sval 2 | Sval 3 |
|---|---|---|---|---|---|---|
| measurements | Simple eBC (MMR) for surface sample | 8000* | 2200 | 5000* | 5000* | 5000* |
| | Sfc eBC (g m$^{-2}$) | 0.04 | 0.01 | 0.05 | 0.08 | 0.12 |
| | eBC sub-surface snow (ng g$^{-1}$) | 200* | 200* | 3750 | 3000 | 2000 |
| | LAP coverage (m$^2$ m$^{-2}$), $A$ in (Eq. 2) | 0.34 | 0.09 | 0.41 | 0.58 | 0.73 |
| albedo estimates | Albedo simple eBC (SNICAR-Online) | 0.64 | 0.73 | 0.67 | 0.67 | 0.67 |
| | Albedo w/o sfc layer (SNICAR-Online) from row 3 sub-surface eBC ($\alpha_s$ in Eq. 1) | 0.80 | 0.80 | 0.69 | 0.71 | 0.73 |
| | Albedo with sfc layer (Eq. 1) | 0.38 | 0.68 | 0.31 | 0.18 | 0.086 |
| | Albedo assuming MAC of 5 for sfc layer | 0.54 | 0.73 | 0.49 | 0.35 | 0.23 |

Table 1: Example data and calculations of albedo based on measurements from Vallunaraju mountain in the Cordillera Blanca, Peru (Vallu 1 and 2) and Svalbard (Sval 1, 2, and 3). The Vallu examples are different interpretations of one measurement where Vallu 1 assumes that the surface sample was collected using the top 1 cm of snow while Vallu 2 adjusts the value of the surface sample assuming that snow was collected from the top 4 cm. Sval 1, 2, and 3 are three scenarios that differ in their assumptions about what the distribution of LAPs in the snowpack might have been given a single sample from the top 10 cm (Kahn et al., 2017). In rows 1 and 3, actual measured values are

indicated with stars and estimates are un-starred. Simple eBC (row 1) is the eBC value determined from the surface measurement; Sfc eBC (row 2) is LAP concentration on the surface in effective g m$^{-2}$; eBC sub-surface snow (row 3) is the LAP concentration within the snow not including the surface layer; and LAP coverage (row 4) is the area covered by LAPs (m$^2$ m$^{-2}$), the surface eBC multiplied by a MAC to determine the actual area covered by the surface layer (with Eq. 2 applied). The last four rows give albedo estimates from four different methods: Albedo simple (row 5) is the SNICAR-Online albedo estimate for the Simple eBC value (data row 1); Albedo w/o sfc layer (row

6) is the SNICAR-Online albedo estimate ($\alpha_s$) for the in snow eBC value (data row 3); Albedo with sfc layer (row 7) is the estimated albedo from Equation 1; and Albedo assuming MAC of 5 (row 8) is a recalculation of the row above but assuming a MAC of 5 m$^2$ g$^{-1}$ rather than the Fullerene soot value of 9 m$^2$ g$^{-1}$ which could be possible due to clumping of LAPs from melt-freeze processes.



Table 2

| | | Exp A | Exp B | Exp C | Exp D |
|---|---|---|---|---|---|
| visible albedo calculated from spectral measurements | Clean snow albedo | 0.73 | 0.91 | 0.83 | 0.89 |
| | Albedo with LAPs on surface | 0.45 | 0.58 | 0.23 | 0.49 |
| | Albedo with LAPs mixed | 0.51 | 0.71 | 0.58 | 0.71 |
| measured LAP values | eBC surface g m$^{-2}$ | 0.015 | 0.004 | 0.029 | e:0.018 |
| | eBC mixed ng g$^{-1}$ (MMR) | 3300 | 485 | 3340 | e:2500 |
| SNICAR | SNICAR-Online broadband albedo | 0.70 | 0.78 | 0.70 | 0.72 |
| albedo reduction | Decrease measured LAPs on surface | 0.28 | 0.32 | 0.50 | 0.40 |
| | Decrease measured LAPs mixed | 0.22 | 0.20 | 0.25 | 0.18 |


Table 2: Results from four experiments exploring the effect the location of LAPs (on or in the snowpack) on albedo. Rows 1-3: visible integrated albedo estimates are calculated from the spectral albedo measurements for the 450-950nm range. The visible albedo values were determined by integrating spectral albedo across the range of the spectrometer. A 5777C Planck curve was used to weight the integration by solar intensity. Rows 4 and 5 are the LAP values in units of eBC determined by LAHM measurements. Row 6 shows the SNICAR-Online

broadband albedo estimated for the row 5 MMR values. Values for experiment D are estimated as the filter was accidentally dropped before LAHM analysis and a substantial portion of the particles were dislodged. Rows 7 and 8 show the reduction in albedo caused by the LAPs compared to the clean snow albedo demonstrating that LAPs on the surface have a larger impact on albedo than when they are mixed into the snow.



Figure 1:

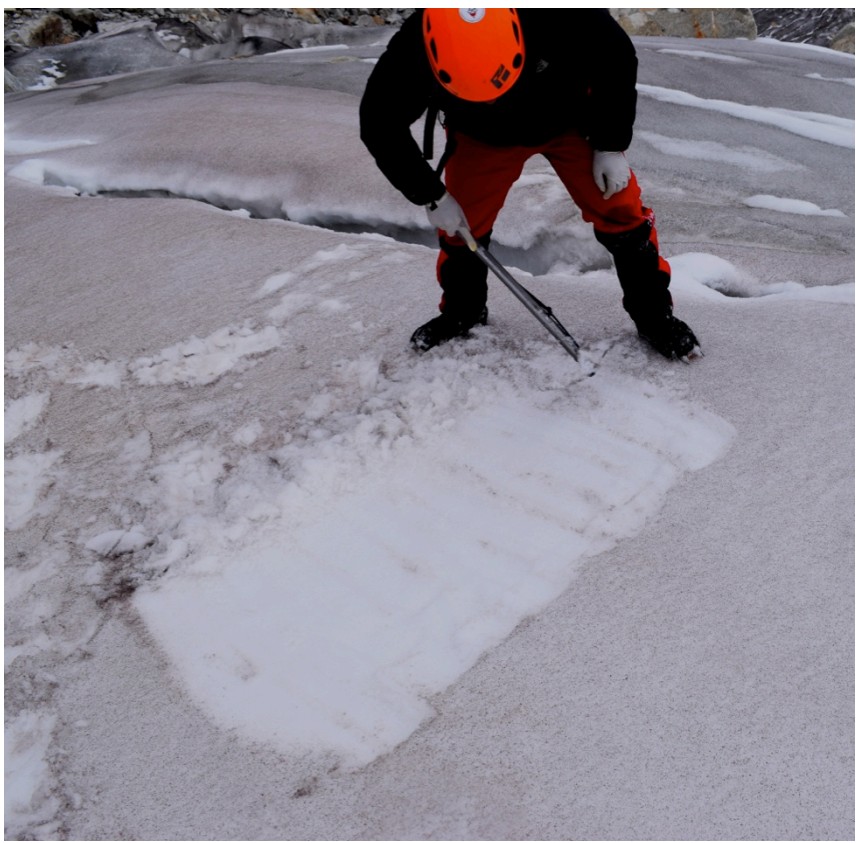

Figure 1: Photo of snow from the ablation zone of a glacier in Cordillera Blanca, Peru in October, 2018 (the end of the typical dry season). This site had likely experienced months of melting without significant fresh snow deposition. The surface of the snow had an obvious layer of LAPs that significantly discolored the snow. Removal of the surface layer revealed much cleaner snow (Sanchez Rodriguez and Schmitt, 375 2018). Photo by Wilmer Sanchez Rodriguez.



Figure 2:

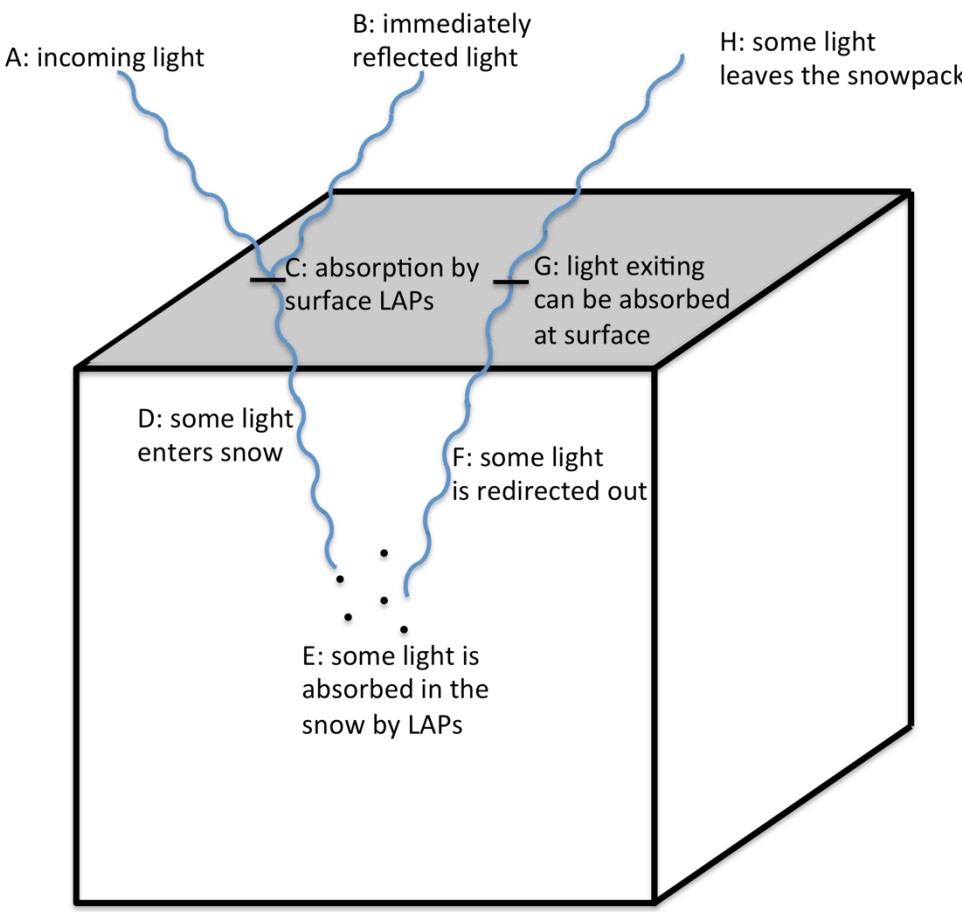

Figure 2: Diagram of light interaction with snow. Incoming light (A) can be immediately reflected (B), absorbed by surface LAPs (at C), or

can enter the snow (D). Once in the snow, light can be absorbed by sub-surface LAPs (E) and some is redirected upward back out of the

snowpack (F). Upon reaching the surface, the light can be absorbed by the surface LAPs (G) or leave the snowpack (H).






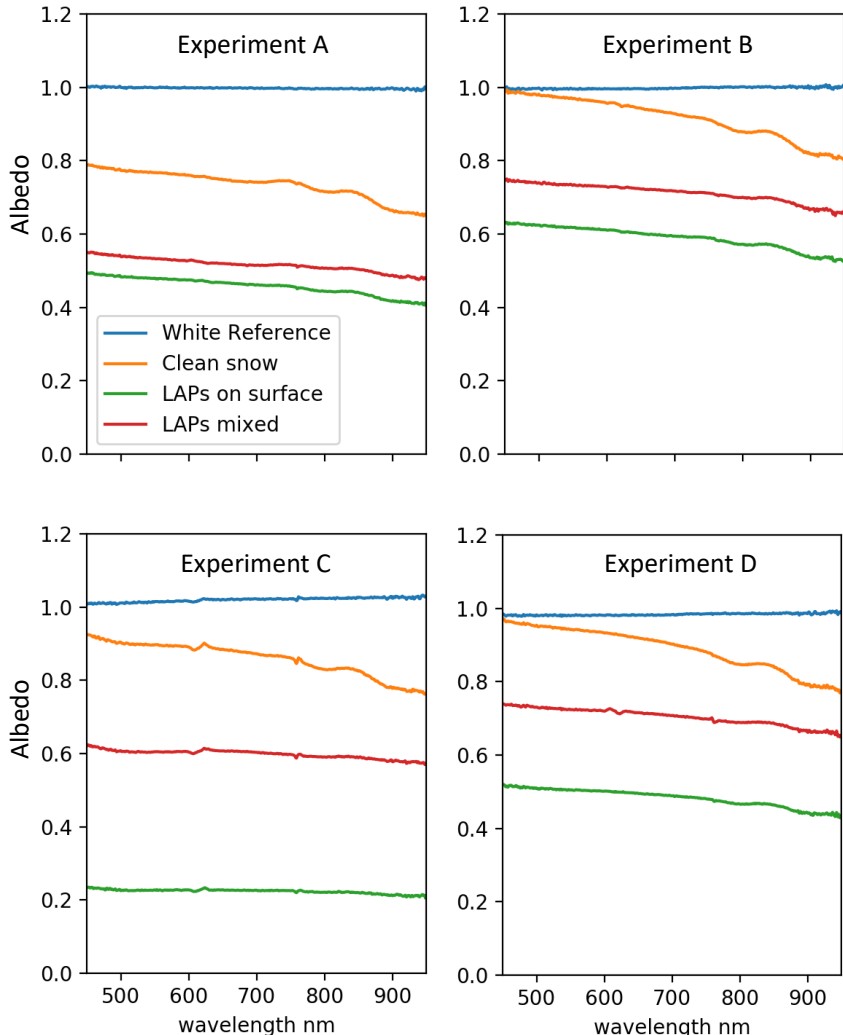

Figure 3: Spectral albedo measurements for experimental examples. The four plots correspond to the four different experiments shown in Table 2. In each plot, from top to bottom the lines indicate the measured spectral albedo for the calibration plate (top, blue), the snow before application of LAPs (second, orange), the albedo with LAPs mixed into the snow (third, green) and the albedo with LAPs on the surface before mixing (lowest, green). Note that the green lines (LAPs on surface) is always lower than the red line (LAPs mixed).