# Peer review of "The measurement and impact of light absorbing particles on snow surfaces"

_The Cryosphere, 2019_

## Referee Comment (RC1) · Quentin Libois (Referee) · 10 Sep 2019

Review of « The measurement and impact of light absorbing particles on snow surfaces », by Carl G. Schmitt et al.

**General comments**

This paper investigates the impact of a thin layer of light absorbing particles (LAP) on the albedo of a snowpack, compared to an equivalent snow layer in whiche the LAP are well-mixed. It provides a theoretical framework to account for such a layer, and applies this framework to snow albedo computations for various snowpacks. An experiment is also set up in the field at a high altitude site in Colorado (USA), which qualitatively corroborates the theoretical findings. Finally, a sampling method is proposed to distinguish well mixed LAP from thin concentrated layers of similar LAP. This study overall demonstrates that the impact of LAP on albedo is much stronger when the latter are concentrated at the top of the snowpack than when they are homogeneously distributed within the snowpack. It means that the vertical resolution at which mass mixing ratios (MMR) measurements of LAP are performed can greatly impact the estimated albedo of a snowpack.

The topic of the study is relevant to *The Cryosphere*. The paper is well written and relatively easy to follow. However the novelty of the research is more questionable because it has been known for a long time than the impact of LAP strongly depends on their location within (or on top of) the snowpack. It has the merit, though, to propose a method for albedo computations in case a layer of LAP is located on top of the snowpack, and it sends a warning to people used to perform LAP measurements in snow, with some suggestion (clearly illustrated on a webpage) for a sampling protocole. The utility of such albedo computations is unfortunately poorly illustrated, which limits the paper impact. The physics behind the albedo computations is also approximative and some critical details regarding spectral measurements and snow physical properties make the study too approximative. Eventually the interpretation of the experiments is very limited. I believe major revisions could strenghthen the impact of the paper and make it singular among an already numerous literature on the topic.

**Specific comments**

1) The abstract is not really an abstract, it is more a condensed introduction. An abstract is meant to provide all the main quantitative results of the study. The abstract should be entirely rephrased to put forward the results and provide enough details, so that a reader would not need read the full paper to catch the essence of it.

2) There seems to be a direct link between LAP vertical distribution and albedo estimations. However, if someone wants to know the albedo of a snowpack it's definitely easier and more accurate to measure it than to measure all the relevant vertical properties of the snowpack to feed a radiative transfer code. Hence it would be very helpful to understand in which context such albedo computations are needed. I think that it is most relevant to estimating the radiative forcing of LAP in snow, and to compute albedo in numerical models for weather or climate predictions (e.g. Tuzet et al., 2017). In general, the study references too few papers which highlights a lack of context.

3) In the past, studies of the impact of LAP on snow albedo have mostly considered MMR, as pointed by the authors. It is not clear what the limit of this representation is, if the topmost layers in such representations become thinner and thinner. Said differently, how do albedo computations with a 1-cm-thick layer containing 8000 ng $g^{-1}$ of eBC differ from those obtained with the introduced surface layer ? How thin should be the topmost layer in the classic MMR representation to match the surface layer value ?

4) The attempt to isolate the LAP surface layer from the snowpack underneath is interesting. However the physics behind this is not very rigorous. First of all, the so-called *surface reflectance,* estimated very simply from the asymmetry parameter of snow, is wrong. The asymmetry parameter *g* of a particle is wrongly defined. It is not the ratio of forward to total scattered radiation, but the mean cosine of the deviation angle between incident and scattered light. In particular, an asymmetry parameter of 0 means that as much light is scattered backward than forward. It does not mean that nothing is scattered forward as suggested by the authors. The paper by Bohren (1987) may provide useful insight to solve this issue. The quantity you defined is more likely to be (1-g)/2. More exactly you could find formulae for single scattering reflectance, e.g. in Khokanovsky (2002). Eventually, this quantity will depend on the solar zenith angle, which is not mentioned at all in the manuscript.

Also, the computation of the total area covered by a given amount of LAP is very approximative. It seems that the LAP is first treated as a dilute medium to compute its MAC, but then a somehow arbitrary (at least not rigorously justified) scaling factor is applied to account for LAP overlaping. This point deserves more explanations, because the impact on the overall albedo is very large, and it is not properly accounted for in the uncertainty analysis. Reaching an albedo accuracy of 0.005 with such a loose definition is unrealistic. Also, it appears quite easy to obtain a total blocking of incident radiation with this definition, while the remaining snow in the LAP layer would still let some light travel through it.

5) At no occasion the physical properties (primarily density and specific surface area SSA) of the snowpack are defined, while they are certainly required by SNICAR. In particular all the quantitative results of the study strongly depends on the SSA, which is not discussed at all. Also, the used configuration of SNICAR is not detailed (number of layers, snowpack thickness, underlying material, solar zenith angle etc.). It's also worth noting that SNICAR assumes spherical particles for snow, while the authors refer to hexagonal crystals to compute surface reflectance, which sounds inconsistent.

6) The spectral dimension of albedo is only very loosely discussed. The wealth of the spectral albedo measurements is unfortunately poorly explored because only broadband albedo values are given. The same study could be done at individual wavelengths before to work on broadband albedo. Because the impact of LAP strongly depends on the wavelengths, such an initial step would provide much more physical insight and could potentially be more convincingly supported by spectral albedo measurements. In particular the light penetration depth of radiation in the snowpack is never mentioned, while it provides a good estimate of where LAP might still impact snow albedo. The authors are definitely invited to discuss this spectral dimension in a future version, and they have room for it.

7) The *in situ* experiment is not sufficiently well described, and the results analysis is definitely too short. Figure 3 shows very distinct features for distinct experiments, that should be analyzed in more details, because they certainly contain unseful physical insight.

**Technical corrections**

l.15 : « deceptive » is subjective and should not be used here

l.15 : « surface accumulation » is not defined

l.17 : the link between sampling strategies and «estimates of albedo » is not explained. It is a lack of context

l.19 : be more quantitative than « top thin layer » which could be 1 mm or 1 cm

l.20 : this sentence is redundant with l.17

l.23 : do your measurements really confirm that the new sampling method enables better albedo estimates ? Be more quantitative anyways

l.33 : provide a reference for the 100 ng g$^{-1}$ value

l.34 : collect → concentrate ?

l.34 : remove last part of this sentence and just keep the reference to the figure

l.37 : This paragraph could be put in first position in the introduction because it explains why LAP are expected in snow. Before to detail how it can be measured and what values are generally encountered. Please provide references to detailed descriptions of the three processes mentioned.

l.42 : remove the fact that it is « uncommon » because it suggests that your study is very marginal, while it is certainly not

l.44 : the « two-dimensional layer » is an awkward term. Valid for the whole manuscript. Use «thin » or « concentrated » layer ?

l.49 : would it be possible in such a model to define a top layer of 1 mm with 0.5 g g$^{-1}$ of eBC ? Would it significantly differ from the layer formulation presented later on ?

l.52 : the end of this sentence is not clear at all. Is there a difference between energy absorbed in the so called top layer and in a 1 mm layer ? Please detail the impact on the snowpack energy budget if relevant

l.62 : this first paragraph looks more like an introduction

l.71 : result → value or estimate

l.74 : it is reflected because it interacts with snow, so this is poorly formulated

l.76 : it can also be absorbed by the material under the snow if the snowpack is thin

Figure 2 : It should be improved to support the corresponding paragraph which is currently hard to follow. I recommend showing a cross section of the snowpack, with a well identified surface layer. That layer should display some « holes » to allow sunlight to directly reach snow and be reflected without interacting with the LAP. The terms $R_s$, $\sigma$ and $\alpha_s$ should appear in this figure.

l.92 : why such an assumption ? Is it relevant or does it cover the literature range on this MAC ?

l.95 : what if the sample had been collected on 3 cm ? The surface layer would have been three times more concentrated in eBC, and the fraction of blocking would have been unity, meaning an albedo of 0. This highlights the limitation of this approach to estimate the properties of the surface layer. It seems very easy to obtain a fully bocking layer, while the small amount of snow, even in a few mm layer, would allow photons to travel and finally be reflected by this topmost layer. As a conclusion, a layer still has some thickness which allows photons to travel between LAPs.

l.98 : does the MAC depend on wavelength ? If so, please precise it, and how do you handle the fact that MAC might vary across the solar spectrum on which broadband albedo is computed ?

l.100 : how do you get this 34 %?

l.115 : Note that SNICAR certainly provides an albedo of 0 for a very high load of BC, which seems contradictory with your assumption that some of the radiation is reflected back (independently of the BC load). Could you clarify this ?

l.115-120 : very hard to follow. Should be rephrased or numbers should be displayed in Fig. 2

l.124 : how thick is the polluted layer in SNICAR simulation ?

l.132 : this equation should come much earlier, after paragraph l. 71-80, and the terms defined at this early stage

l.133 : the $R_s$ term is very questionable, because it is valid only if directly reflected radiation by snow is not absorbed by surrounding LAPS, which is unlikely if the snow is surrounded by LAP.

l.135 : $R_s$ corresponds to clean snow, while practically snow is not necessarily clean. Isn't it inconsistent ?

l.139 : this definition is wrong as explained previously

l.139 : grain size was not defined nor quantified up to now…

l.141-146 : where does it come from ? Should be rigorously derived

l.143 : « it was found » → provide details or reference or plots

l.150 : need more details about spectral range, resolution, instrument etc. Why do you mention albedo measurements here while they do not appear in Table 1 ?

l.151 : « reduced » compared to what ? Spectral albedo or any other albedo ? Could you define HDRF so that the reader can interpret the value of 0.16. Are you sure that the quantity measured (HDRF) can be compared to the albdo defined in Eq. 1 ? Which seems to be some diffuse directional-hemispherical albedo ? This should by the way be detailed.

l.170 : it is not clear what the authors aim to do. An uncertainty analysis should be perfomed forward, starting from the various contributions, and ending up with an uncertainty on computed albedo. Here this is not an uncertainty analysis that is performed. Such analysis would be very complicated. Just say that you investigfate how much of BC in a surface layer does result in a change of albedo of 0.005. Note that 0.005 is certainly far more accurate than your albedo measurements.

l.174 : again the surface reflectance is wrongly defined

l.178 : what about these properties ? What values are taken, what is the range of possible values and the impact on the overall albedo ?

l.179 : you do not prove that the impact of well-mixed impurities is less than neglecting the surface layer, you just state it.

l.182 : « we were compelled » → too subjective wording

l.183 : what instrument for spectral albedo measurements ?

l.186 : instrument details arrive too late and are not complete. What light collector is used ? What field of view, spectral range etc.

l.189 : what is a calibration spectrum ?

l.195 : how can this estimate of 3 cm be reliable ? It seems extremely hard to assess in the field

l.200 : how do the 4 experiments differ ? Is the initial snowpack similar ? Were they all taken the same day ?

l.214 : Could you discuss more the change in albedo reduction because sometimes mixing LAPs does not change by two the albedo reduction. In general, please describe in more details the differences obtained among these 4 experiments.

Figure 3 : why is the albedo of clean snow at 450 nm so low for experiment A ?

l.216 : why don't you use a solar spectrum instead of a Planck curve ?

l.216 : you apparently compare a visible to a broadband albedo, this is inconsistent

l.220-221 : remove this last sentence which is redundant with previous text

l.232 : scraping surface snow does not sound like a robust and reproductible protocol. It's very qualitative. Is it really critical if you take too much surface snow, given you can correct for the contribution of subsurface snow later on ?

l.246 : showing the equations for these calculations would be helpful

l.254 : in general this conclusion is not well written and does not summarize the main results. Also it provides too few perspectives.

l.260 : it is still not clear why albedo should be estimated from such measurements (which should again include snow physical properties) rather than measured

l.261 : using « Recommendations » here is awkward. You should provide some major results before to present the sampling method

l.268: I doubt brushing fresh snow without getting underneath LAPs is an easy task.

l.270 : « where the lead author lives » is not very useful for the readers

**References**

Bohren, C. F. (1987). Multiple scattering of light and some of its observable consequences. *American Journal of Physics, 55*(6), 524-533.

Kokhanovsky, A. A. (2002). Analytical solutions of multiple light scattering problems: a review. *Measurement Science and Technology, 13*(3), 233.

Tuzet, F., Dumont, M., Lafaysse, M., Picard, G., Arnaud, L., Voisin, D., ... & Morin, S. (2017). A multilayer physically based snowpack model simulating direct and indirect radiative impacts of light-absorbing impurities in snow. *The Cryosphere, 11*(6), 2633-2653.

---

## Referee Comment (RC2) · Anonymous Referee #2 · 12 Sep 2019

The paper focuses on the impact of the vertical distribution of light absorbing particles (LAPs) on versus in snowpacks on surface albedo, and it specifically addresses how sampling must be designed to account for cases where LAP are on the surface of the snow, rather than mixed into the snow. A sampling strategy is recommended. A qualitative demonstration of the effect on albedo of having LAP on the snow surface versus mixed into the snow.

The paper raises a relevant and, in for some snowpacks, important point: Albedo calculations that use mass mixing ratios (MMRs) of LAPs in snow which assume the LAPs are uniformly mixed into the snowpack will be biased toward higher albedo, if in fact the LAP are concentrated at the snow surface. This topic is relevant for The Cryosphere, where many paper on the effects of LAPs on snow albedo have been published.

[Figure]

However, I have several significant difficulties with the paper:

1) The authors present their proposed sampling strategy as if it is a \*new\* strategy. In fact it's simply a refinement of the approach used in previous studies. While some studies have analyzed for LAP using a single or uniform sampling depth at all locations, in some cases sampling has specifically tried to isolate layers that appear to be uniformly mixed – including, sometimes, sampling very thin surface layers (e.g. ice crusts), and in some cases sampling both 'surface' and 'sub-surface sample' layers, even in some cases with the surface sample covering quite a thin top layer of the snowpack. What is suggested here is a sampling strategy is simply taking the same overall approach, but for cases where there is actually an accumulation of LAP on the snow surface of sufficient thickness limiting the surface sample to an even thinner layer (e.g the top couple mm of the snow surface).

2) This surface layer of LAP is alternately presented as being a "2D" layer and being sampled over some depth – so it is clearly not really only 2-dimensional. The text should be edited for better consistency and accuracy. While I understand the gist of what the authors are trying to say, the surface LAP always has some finite depth.

3) I believe that the suggested method for calculating surface albedo includes an incorrect interpretation of the asymmetry parameter, g. A g of zero, for example, means that light is scattered equally in the forward and backward directions (the case for very small particles). A g of 0.80 does not mean that 20% of the light is scattered upwards.

4) What SZA is used in these calculations?

5) It's not clear how the light-scattering properties of the particles on the snow surface are accounted for; as far as I can tell only absorption is accounted for. By process "B" in Figure 2, some amount of incoming light is "immediately reflected"– but in the text, the light lost before reaching the snowpack is all via absorption (lines 98-100). Reflection by the snow, not the particles, is then accounted for using an approximate asymmetry parameter (lines 104-106; correction needed per point 3 above). I don't see discussed

anywhere how much scattering there would be off of the particles themselves. LAPs in ambient snow will rarely if ever be pure soot. In the case of glaciers or snowpacks near light-colored deserts, for example, particles accumulated on the snow surface could be much more strongly light scattering than soot. This would affect both the up-scattering/reflection of incoming sunlight and how much light propagating up through the snowpack (path "F" in Figure 2) is then scattered back down into the snowpack by the surface particle layer, where it will have additional opportunities to be absorbed by LAP mixed in the snowpack. As such, I believe the model given accounts only for surface layers of particles that are nearly purely light absorbing, and this is not valid for use with ambient snowpacks.

6) The suggested sampling strategy is to collect one sample that isolates as best as possible the surface layer of particles, while collecting as little of the surface snow as possible, then collecting a single sub-surface sample (depth not specified). However, vertical variations in the MMR of LAPs in the sub-surface snow may also be important. For example, a snowpack that has significant accumulation of particles on the surface might also have a thin melt- or wind-crust that has quite a different LAP MMR than the snow immediately below that. Or perhaps there was a snowfall of a few cm, then a long period of no new snow but significant dry deposition, then another cm of relatively clean new snow – then the accumulation of particles on that snow surface. While vertical variations in the MMR of LAP in the snow will be most important near the snow surface, all such vertical variations will matter down to the penetration depth of sunlight (approx. 10cm, +/- depending on LAP concentrations and snow grain size). The proposed sampling strategy is certainly a good *minimum* requirement for snow with very high concentration of LAPs on top of a snowpack. However, a single sub-surface sample as a recommended strategy is only appropriate where the snow below these particles has very uniform LAP MMR – a very specific configuration that will occur in only a very limited number of cases. If an improved sampling strategy is to be proposed, it would be better to account for a larger range of possible cases by specifying that, whenever possible to do so, multiple sub-surface samples should be
collected. If visible layers are present in the snow, sample depths could be determined visually; where no visible vertical variations are present, if several sub-surface layers can be collected it would be best to sample thinner layers towards the snow surface (e.g. collecting samples of 2cm depth, 3cm deep, then 5cm deep for the case where three sub-surface layers could be collected, to cover the full 10cm that light is likely to penetrate).

7) A simple thought experiment is sufficient to know it must be the case that concentrating LAPs at the surface of the snowpack will have a greater effect on albedo than will mixing the same amount of LAPs throughout some depth of the snowpack. This does not need to be qualitatively demonstrated via experiment. What *is* needed is quantitative verification of the proposed theoretical method of calculating albedo in the case where LAPs are on the snow surface, using the sampling strategy and calculation method proposed. The experiment described in Section 3 (results shown in Figure 3) doesn't provide such a quantitative test. Thus, I question the value of including discussion of this experiment in the paper.

8) If the experimental results are going to be included two things need to be addressed:

a) The setup had the FieldSpec located 30cm above the snowpack, viewing an area 20-30cm in diameter to which LAPs had been added. I am assuming the FieldSpec has something like a cosine-weighted input. (Such details should be specified). The instrument therefore would have been viewing an area that includes snow without added LAP, correct? This is not mentioned or accounted for.

b) The "clean snow" (snow to which LAP has not been added) is reported as having an eBC MMR of 12 ng/g. In the four cases shown in Figure 3, the ~500nm albedo of this snow varies from <0.8 to >0.95. An MMR of 12 ng BC/g snow isn't sufficient to explain a 500nm snow albedo of <0.8. In addition there's either actual variations in the snowpack or significant measurement uncertainties that are not being accounted for here, demonstrating that this is a very qualitative analysis.

---

## Referee Comment (RC3) · Anonymous Referee #3 · 27 Sep 2019

The manuscript makes the point that differentiating layers of light absorbing particle (LAP) mass mixing ratios (MMR), particularly the surface layer, is necessary to derive a realistic value for the resulting surface albedo. A theoretical justification is provided which is underlined by a qualitative field experiment. Based on this, a sampling protocol is suggested and instructive online material is provided.

The manuscript raises the important point that bulk LAP MMR in snow samples can underestimate the MMR in the surface layer where LAPs can accumulate due to melting snow, sublimation or dry deposition of LAPs. The surface layer is obviously the most important layer for the resulting surface albedo. It is hence important to characterize the surface layer separately and harmonize the sampling protocol in the community to produce more comparable results. While this issue certainly deserves dissemination

and is a topic for 'The Cryosphere', the manuscript does not present any new results per se, but is rather method focused. This makes it perhaps better suited for a 'Technical Note', rather than a full scientific manuscript, especially because the presented field experiment results are very qualitative. In addition, several points will have to be addressed to correct and complete the work, hence I suggest major revisions.

General comments:

1) The manuscript will greatly benefit from more context: Why is the accurate determination of snow surface albedo important? In which models and how are such measurements used? What is the use of a point and snapshot measurement as described in this work? A common challenge is the upscaling spatially and temporally with such field measurements. Spatial heterogeneity of LAP MMR in surface snow is one issue. The other aspect is, which has also not been discussed, that LAP do not only accumulate at the surface under certain conditions, but that also new snowfall can reduce the MMR of LAP at the surface temporarily. So the temporal dimension is also very important to consider as high and low LAP MMR conditions at the surface might alternate. To be able to model this a number of processes need to be characterized towards their importance for surface albedo, and it will be important to convey where in this more complex consideration the value of this work lies. There is a vast amount of literature and some of it should be reflected in the introduction.

2) This work is entitled "The measurement and impact of light absorbing particles on snow surface", however it focuses on the 'measurements' rather than the 'impact' (see need for additional context above). In addition, the work focuses on black carbon (BC) rather than the diverse types or mix of LAP to which mineral dust, brown carbon, organic carbon, microorganisms and others belong as well. Measuring all of these will further complicate albedo determination and is beyond the scope of this manuscript. But the mix is an important feature of real snow samples and will vary in composition by location and season. Hence, either a discussion of how this could be addressed and how sensitive albedo calculations are to it is needed in the manuscript, or at the

very least pointing out the issue is needed, and the title should be changed to "The measurement of black carbon in snow surfaces".

3) Some more precise definition of the "2D surface layer" is needed. Since it practically cannot be 2D more details on the depth in this particular study need to be included and it also should be discussed how the depth of the surface layer can vary by location. There might be examples where the LAP enrichment happened within millimeters, while for other locations a centimeter depth would describe the surface layer equally well. Also, there should be more information given on the depth to which light penetrates the snow and on which parameters this depends (snow characteristics, LAP MMR etc.), to provide a reasonable range of the overall sampling depth that should be covered in different layers. One assumption this work makes is that under the surface layer LAPs are mixed homogeneously, which most likely is a special case. In most instances, more layers will need to be defined. This is briefly mentioned at the end of the manuscript.

4) The definition of the asymmetry parameter 'g' is not correct, please revise.

5) The albedo calculation with SNICAR requires information on snow grain size, a critical parameter for the snow albedo (in each of the layers). Not only the spread and MMR of LAP at the surface are important, but also the changing snow grain size during the LAP accumulation process. There is no information in the manuscript about the measurements or assumptions made. Please add this. The same is true for the solar zenith angle.

6) The optical properties of BC depend on its size. There is literature (e.g. Schwartz et al., 2013, DOI: 10.1038/srep01356) showing that BC size in snow is larger than atmospheric BC due to (post-)deposition processes. This shifts the MAC to smaller values. This process is particularly relevant for situations where BC accumulates at the snow surface due to melting or sublimation, or when the MMR is so high that particles stick to each other. This work does not take the size of BC into account and also does

not discuss how the artificial soot doping of the snow in the section 3 experiments compares to atmospheric soot that is deposited. While it is evident that this information cannot be retrieved anymore and is also not trivial to derive, at least a discussion needs to be included because the effect partly compensate the estimated albedo reduction reported here.

7) The paper would benefit strongly from a more comprehensive uncertainty analysis that includes for example variation of the following parameters: snow grain size, MAC value, estimate of the LAP covered area (because it is not clear based on which consideration the overlap is estimated in the manuscript, should be made clear), the surface layer thickness etc. This will be very instructive to learn where the largest uncertainties come from, and for the sampling community for where to pay attention for most accurate measurements. The would particularly benefit section 3, which is qualitative only, and hence does not add new information to the manuscript per se. The information section 3 provides is also provided in section 2, the theoretical consideration.

8) Figure 2 should be made more accurate. The variables mentioned in the text should appear, the transmitted and reflected radiation should be marked by arrows. Snow grains and layers are important features as well as an indication of the light penetrating depth.

9) The suggested sampling strategy is not new per se, it is rather a refinement of an existing method.

Specific comments.

l. 39f: Add to point 2) that particularly in areas with open soil or mineral dust wind blown addition of LAPs can be important.

l. 42: Are these situations really uncommon? Why not simply state that they occur?

l. 90: The authors do not use four different methods, they rather use four different model variants of SNICAR.

l. 101: from the text it is not apparent why the perfect spread of LAPs would result in 0.36 m2/m2. Please add an explanation.

l. 143: What does "it was found" mean. Is there a reference for it? If yes, it needs to be added. If the authors found it should state "we found".

If the Schmitt et al., 2019, AMTD manuscript is available it should be added. If not, citing it does not help very much.

Figure 1: The caption states that it is the ablation zone that has not experienced any significant snowfall in the dry season. In that case, I would not expect snow on the ice of the ablation zone. And if there are remnants would they not have undergone several melting and freezing cycles? If that is the case what was the SNICAR input for snow grain size? Or is the dry season the cold season and the snow does not melt? This needs to be clarified.

Technical comments:

l. 38: replace "through" by "when"

l. 62: "..., an MMR is a reasonable..."

l. 86: "In the Svalbard case, ..."

l. 89: ".. those scenarios to estimate the range..."

l. 91: delete "using" in front of "SNICAR-Online"

l. 182: reformulate to "... LAP concentration, we experimentally validated these results."

l. 266: What is meant by "and if a surface layer is possible"? Please rephrase.

---

## Author Comment (AC1) · 13 Jan 2020

The authors would like to thank the three anonymous reviewers for their comments and suggestions. In the revised manuscript, we will address all of these comments. Detailed responses to how these comments will be addressed are included in this document. In reading the reviews, there were a few topics that we feel require fundamental (if minor) changes to the manuscript to increase understanding. We will address those here and refer to these comments later when responding to individual comments.

The terminology of a "layer" in the manuscript will be changed in the revised manuscript. We regularly referred to "surface layers" of LAPs in the manuscript, but it seems easy to confuse this with different "layers" in the snow. The difference being that "layers" in the snow are generally assumed to have macroscopic thicknesses on the order of centimeters while a "surface layer" of LAPs in the manuscript refers to a layer of non-ice particles, that are not mixed into a macroscopic snow layer. In the revised manuscript the LAP layer will be referred to as a "coating" or "surface coating" of LAPs to distinguish it from snow layers including layers of snow with LAPs mixed in. We feel that this change will make the discussion and interpretation more clear.

Below is a new table, a version of which will go in the revised manuscript. The Vallunaraju case is used to demonstrate the impact of different layer thicknesses. SNICAR was run with the same total eBC with the eBC being distributed evenly through different layer thicknesses (MMR increased as layer thickness decreased conserving total eBC). The visible albedo calculated by SNICAR (assuming a total snowpack depth of 10 meters) shows large differences depending on how one partitions the LAPs. Note that the modeled albedo using equation 1 in the manuscript is 0.38. Also, note that this was a much more extreme case than that shown in Figure 1, thus, although we don't have direct albedo measurements from the case, we suspect that the actual albedo was closer to 0.3 than 0.7. This table will be included in the manuscript as well as discussion of these results.

| layer thickness (m) | LAP conc (ng/g) | LAP below | visible albedo |
|---|---|---|---|
| 0.1 | 800 | 0 | 0.8914 |
| 0.01 | 8000 | 0 | 0.7013 |
| 0.001 | 80000 | 0 | 0.4189 |
| 0.0001 | 800000 | 0 | 0.3116 |
| 0.00001 | 8000000 | 0 | 0.2974 |
| 0.000001 | 80000000 | 0 | 0.2983 |

All three reviewers mention the use of the asymmetry parameter is erroneous and incorrectly defined. In the revised manuscript, the surface reflection will be redone more realistically. The 15% estimate in the original manuscript is likely too high based on initial calculations suggesting that this factor is less important.

**Anonymous Referee #1**

Review of « The measurement and impact of light absorbing particles on snow surfaces », by Carl G. Schmitt et al.

**General comments**

This paper investigates the impact of a thin layer of light absorbing particles (LAP) on the albedo of a snowpack, compared to an equivalent snow layer in whiche the LAP are well-mixed. It provides a theoretical framework to account for such a layer, and applies this framework to snow albedo computations for various snowpacks. An experiment is also set up in the field at a high altitude site in Colorado (USA), which qualitatively corroborates the theoretical findings. Finally, a sampling method is proposed to distinguish well mixed LAP from thin concentrated layers of similar LAP. This study overall demonstrates that the impact of LAP on albedo is much stronger when the latter are concentrated at the top of the snowpack than when they are homogeneously distributed within the snowpack. It means that the vertical resolution at which mass mixing ratios (MMR) measurements of LAP are performed can greatly impact the estimated albedo of a snowpack.

The topic of the study is relevant to *The Cryosphere*. The paper is well written and relatively easy to follow. However the novelty of the research is more questionable because it has been known for a long time than the impact of LAP strongly depends on their location within (or on top of) the snowpack. It has the merit, though, to propose a method for albedo computations in case a layer of LAP is located on top of the snowpack, and it sends a warning to people used to perform LAP measurements in snow, with some suggestion (clearly illustrated on a webpage) for a sampling protocole. The utility of such albedo computations is unfortunately poorly illustrated, which limits the paper impact. The physics behind the albedo computations is also approximative and some critical details regarding spectral measurements and snow physical properties make the study too approximative. Eventually the interpretation of the experiments is very limited. I believe major revisions could strenghthen the impact of the paper and make it singular among an already numerous literature on the topic.

**Specific comments**

1) The abstract is not really an abstract, it is more a condensed introduction. An abstract is meant to provide all the main quantitative results of the study. The abstract should be entirely rephrased to put forward the results and provide enough details, so that a reader would not need read the full paper to catch the essence of it.

The abstract will be re-worded in the revised manuscript

2) There seems to be a direct link between LAP vertical distribution and albedo estimations. However, if someone wants to know the albedo of a snowpack it's definitely easier and more accurate to measure it than to measure all the relevant vertical properties of the snowpack to feed a radiative transfer code. Hence it would be very helpful to understand in which context such albedo computations are needed. I think that it is most relevant to estimating the radiative forcing of LAP in snow, and to compute albedo in numerical models for weather or climate predictions (e.g. Tuzet et al., 2017). In general, the study references too few papers which highlights a lack of context.

The revised manuscript introduction will include more details on uses of this type of research. Specifically, dry deposition can be estimated by modeling of the airflow of plumes of smoke or other LAPs.  In many cases, it is not possible to measure the albedo directly.  Also, since LAP concentrations evolve with time during the melt season, correctly estimating surface snow melt (and deposition of the melted snow's LAP load) can improve the forecasting of albedo.

3) In the past, studies of the impact of LAP on snow albedo have mostly considered MMR, as pointed by the authors. It is not clear what the limit of this representation is, if the topmost layers in such representations become thinner and thinner. Said differently, how do albedo computations with a 1-cm-thick layer containing 8000 ng g$^{-1}$ of eBC differ from those obtained with the introduced surface layer ?

How thin should be the topmost layer in the classic MMR representation to match the surface layer value ?

This is a very interesting question that has driven to a large extent the revision of the manuscript. SNICAR has been run with two layers, the top layer containing all of the "surface coating" LAP mass (in SNICAR, the MMR was increased with each decrease of layer thickness so that the total LAP mass remained in the top layer), and a 10 meter layer below with low LAP concentrations. With an initial top layer thickness of 10 cm, the calculated albedo was much higher than predicted by the equations in the submitted manuscript, but as the layer was made thinner, the snowpack albedo leveled off at a value similar to the value predicted by the submitted manuscript equations. The maximum layer thickness that resulted in similar results was 50 microns. A new table will be included in the revised manuscript to show these results and the SNICAR results calculated with different layer thicknesses.

4) The attempt to isolate the LAP surface layer from the snowpack underneath is interesting. However the physics behind this is not very rigorous. First of all, the so-called *surface reflectance*, estimated very simply from the asymmetry parameter of snow, is wrong. The asymmetry parameter *g* of a particle is wrongly defined. It is not the ratio of forward to total scattered radiation, but the mean cosine of the deviation angle between incident and scattered light. In particular, an asymmetry parameter of 0 means that as much light is scattered backward than forward. It does not mean that nothing is scattered forward as suggested by the authors. The paper by Bohren (1987) may provide useful insight to solve this issue. The quantity you defined is more likely to be (1-g)/2. More exactly you could find formulae for single scattering reflectance, e.g. in Khokanovsky (2002). Eventually, this quantity will depend on the solar zenith angle, which is not mentioned at all in the manuscript.

As stated in the initial comments, the surface reflection will be redone and will likely come out lower thus reducing the impact of this factor.

Also, the computation of the total area covered by a given amount of LAP is very approximative. It seems that the LAP is first treated as a dilute medium to compute its MAC, but then a somehow arbitrary (at least not rigorously justified) scaling factor is applied to account for LAP overlaping. This point deserves more explanations, because the impact on the overall albedo is very large, and it is not properly accounted for in the uncertainty analysis. Reaching an albedo accuracy of 0.005 with such a loose definition is unrealistic. Also, it appears quite easy to obtain a total blocking of incident radiation with this definition, while the remaining snow in the LAP layer would still let some light travel through it.

The MAC is a physical property of the substance being measured in units of "area of effective absorption" per "unit of particles" where "unit of particles" can be mass or number or something else to define a quantity. With that, we can define a total absorptive area, but in some cases, the total absorptive area could be higher than the total area being investigated. This suggests that some of the area could be hidden, and the assumption is that the absorptive area of multiple particles could potentially overlap and since an absorptive area cannot absorb more light than is coincident, overlapping is likely. To define the area covered, a python program was written to estimate overlapping by randomly distributed particles. Simply, a 1000 by 1000 array was created to represent an "area" of snow, then "particles" were assigned to points at random within the array. Any point in the array that was occupied by one *or more* particles was considered to be occupied. If, randomly, no particles were assigned to a point, then the point was assumed to be clean. Using this strategy, numerous runs of the program were made adding more and more particles. A curve was fit to the results which is parameterized in equation 2. In the revised manuscript, this will be better explained and a plot will be added showing the parameterization. Regarding your comment that it is "quite easy to obtain total blocking", I am not sure I understand. A

number divided by the same number plus a little (the square root of the number squared plus 1) is always going to be less than 1.

5) At no occasion the physical properties (primarily density and specific surface area SSA) of the snowpack are defined, while they are certainly required by SNICAR. In particular all the quantitative results of the study strongly depends on the SSA, which is not discussed at all. Also, the used configuration of SNICAR is not detailed (number of layers, snowpack thickness, underlying material, solar zenith angle etc.). It's also worth noting that SNICAR assumes spherical particles for snow, while the authors refer to hexagonal crystals to compute surface reflectance, which sounds inconsistent.

The impact of different physical properties of the snow is discussed briefly around line 175.  While the physical properties of the snow are important to estimate albedo using the SNICAR model, substantially delving into this would detract from the message that it is important to consider a surface coating.  The physical properties of the snow do not substantially affect the coating on the surface.  In the revised manuscript, the physical properties used for all the calculations will be listed.

6) The spectral dimension of albedo is only very loosely discussed. The wealth of the spectral albedo measurements is unfortunately poorly explored because only broadband albedo values are given. The same study could be done at individual wavelengths before to work on broadband albedo. Because the impact of LAP strongly depends on the wavelengths, such an initial step would provide much more physical insight and could potentially be more convincingly supported by spectral albedo measurements. In particular the light penetration depth of radiation in the snowpack is never mentioned, while it provides a good estimate of where LAP might still impact snow albedo. The authors are definitely invited to discuss this spectral dimension in a future version, and they have room for it.

It would be very interesting to look at the spectral variability of this effect as it could have significant implications when considering LAPs with variable absorption such as dust and brown carbon. Unfortunately, due to likely time required to advance the study in this direction, we are unable to move in this direction at this time.  We would be very happy to see an investigation on this topic, but cannot conduct it ourselves within the time constraints.

7) The *in situ* experiment is not sufficiently well described, and the results analysis is definitely too short. Figure 3 shows very distinct features for distinct experiments, that should be analyzed in more details, because they certainly contain unseful physical insight.

Due to conditions and time constraints, it was not possible to do the in situ experiments in a well quantifiable way.  As pointed out by another reviewer, the albedo at 450 nm for experiment A was much lower than would be expected and much lower than the others.  The reason for this could not be quantified as the results were calculated days after the experiments.  There was always several minutes as well as changing of positions by the researchers between the plate measurement and the measurement of the LAP doped snow, which could have caused a number of factors affecting the results.  The doped snow measurements were always taken in quick succession with little repositioning of the researchers.  As such we did not feel that trying to exactly quantify the uncertainties would be reasonable but we do believe that the consistency of the observed trends were such that subjectively, they added evidence to the story. Given the difficulty in exactly quantifying the results, we would consider moving this section to supplementary material rather than it being a section in the publication should the editor or reviewers think that would be better.

---

## Author Comment (AC2) · 13 Jan 2020

The authors would like to thank the three anonymous reviewers for their comments and suggestions. In the revised manuscript, we will address all of these comments. Detailed responses to how these comments will be addressed are included in this document. In reading the reviews, there were a few topics that we feel require fundamental (if minor) changes to the manuscript to increase understanding. We will address those here and refer to these comments later when responding to individual comments.

The terminology of a "layer" in the manuscript will be changed in the revised manuscript. We regularly referred to "surface layers" of LAPs in the manuscript, but it seems easy to confuse this with different "layers" in the snow. The difference being that "layers" in the snow are generally assumed to have macroscopic thicknesses on the order of centimeters while a "surface layer" of LAPs in the manuscript refers to a layer of non-ice particles, that are not mixed into a macroscopic snow layer. In the revised manuscript the LAP layer will be referred to as a "coating" or "surface coating" of LAPs to distinguish it from snow layers including layers of snow with LAPs mixed in. We feel that this change will make the discussion and interpretation more clear.

Below is a new table, a version of which will go in the revised manuscript. The Vallunaraju case is used to demonstrate the impact of different layer thicknesses. SNICAR was run with the same total eBC with the eBC being distributed evenly through different layer thicknesses (MMR increased as layer thickness decreased conserving total eBC). The visible albedo calculated by SNICAR (assuming a total snowpack depth of 10 meters) shows large differences depending on how one partitions the LAPs. Note that the modeled albedo using equation 1 in the manuscript is 0.38. Also, note that this was a much more extreme case than that shown in Figure 1, thus, although we don't have direct albedo measurements from the case, we suspect that the actual albedo was closer to 0.3 than 0.7. This table will be included in the manuscript as well as discussion of these results.

| layer thickness (m) | LAP conc (ng/g) | LAP below | visible albedo |
| --- | --- | --- | --- |
| 0.1 | 800 | 0 | 0.8914 |
| 0.01 | 8000 | 0 | 0.7013 |
| 0.001 | 80000 | 0 | 0.4189 |
| 0.0001 | 800000 | 0 | 0.3116 |
| 0.00001 | 8000000 | 0 | 0.2974 |
| 0.000001 | 80000000 | 0 | 0.2983 |

All three reviewers mention the use of the asymmetry parameter is erroneous and incorrectly defined. In the revised manuscript, the surface reflection will be redone more realistically. The 15% estimate in the original manuscript is likely too high based on initial calculations suggesting that this factor is less important.

**Anonymous Referee #2**

The paper focuses on the impact of the vertical distribution of light absorbing particles (LAPs) on versus in snowpacks on surface albedo, and it specifically addresses how sampling must be designed to account

for cases where LAP are on the surface of the snow, rather than mixed into the snow. A sampling strategy is recommended. A qualitative demonstration of the effect on albedo of having LAP on the snow surface versus mixed into the snow.

The paper raises a relevant and, in for some snowpacks, important point: Albedo calculations that use mass mixing ratios (MMRs) of LAPs in snow which assume the LAPs are uniformly mixed into the snowpack will be biased toward higher albedo, if in fact the LAP are concentrated at the snow surface. This topic is relevant for The Cryosphere, where many paper on the effects of LAPs on snow albedo have been published.

However, I have several significant difficulties with the paper:

1) The authors present their proposed sampling strategy as if it is a *new* strategy. In fact it's simply a refinement of the approach used in previous studies. While some studies have analyzed for LAP using a single or uniform sampling depth at all locations, in some cases sampling has specifically tried to isolate layers that appear to be uniformly mixed – including, sometimes, sampling very thin surface layers (e.g. ice crusts), and in some cases sampling both 'surface' and 'sub-surface sample' layers, even in some cases with the surface sample covering quite a thin top layer of the snowpack. What is suggested here is a sampling strategy is simply taking the same overall approach, but for cases where there is actually an accumulation of LAP on the snow surface of sufficient thickness limiting the surface sample to an even thinner layer (e.g the top couple mm of the snow surface).

While it is indeed not a "new" strategy and is a refinement of typical sampling practices, when a significant surface coating is present, the albedo calculations can be markedly different. Numerical experiments with the SNICAR model (done in response to a question from Reviewer #1) lead us to conclude that when a surface coating is present, treating the coating as being 50 microns thick will lead to albedo calculations that converge with the calculation technique described in this publication. While trying to collect only the top 50 microns of snow is impossible, radiatively, all of the mass in the top layer of sampled snow could be modeled to be in a 50 micron surface snow layer. The text of the revised manuscript will detail these SNICAR experiments and the data will be placed in a table.

2) This surface layer of LAP is alternately presented as being a "2D" layer and being sampled over some depth – so it is clearly not really only 2-dimensional. The text should be edited for better consistency and accuracy. While I understand the gist of what the authors are trying to say, the surface LAP always has some finite depth.

True, it is not an actual 2D layer. As stated above, it can be treated as a 50 micron layer. When considering scale, when dealing with ice crystals on the tens of microns scale at the smallest, LAPs can be much smaller, from tens of nanometers for black carbon to a few microns for dust, so a coating of these particles would be very thin compared to ice crystals in the snow pack. Also, surface LAP layers tend to be more common in melting conditions when the ice crystals are larger. The text of the revised manuscript will fully elucidate this.

3) I believe that the suggested method for calculating surface albedo includes an in- correct interpretation of the asymmetry parameter, g. A g of zero, for example, means that light is scattered equally in the forward and backward directions (the case for very small particles). A g of 0.80 does not mean that 20% of the light is scattered upwards.

As stated in the initial comments, the surface reflection will be redone and will likely come out lower thus reducing the impact of this factor.

4) What SZA is used in these calculations?

All input parameters will be detailed in the revised manuscript. Although, as detailed in response to Reviewer #1, the impact of the LAP surface coating is the focus of this publication and the impact is treated similarly regardless of underlying snow properties, so we feel that going into too much detail on the snow properties would reduce the impact of the message we want to get across.

5) It's not clear how the light-scattering properties of the particles on the snow surface are accounted for; as far as I can tell only absorption is accounted for. By process "B" in Figure 2, some amount of incoming light is "immediately reflected"– but in the text, the light lost before reaching the snowpack is all via absorption (lines 98-100). Reflection by the snow, not the particles, is then accounted for using an approximate asymmetry parameter (lines 104-106; correction needed per point 3 above). I don't see discussed anywhere how much scattering there would be off of the particles themselves. LAPs in ambient snow will rarely if ever be pure soot. In the case of glaciers or snowpacks near light-colored deserts, for example, particles accumulated on the snow surface could be much more strongly light scattering than soot. This would affect both the up- scattering/reflection of incoming sunlight and how much light propagating up through the snowpack (path "F" in Figure 2) is then scattered back down into the snowpack by the surface particle layer, where it will have additional opportunities to be absorbed by LAP mixed in the snowpack. As such, I believe the model given accounts only for surface layers of particles that are nearly purely light absorbing, and this is not valid for use with ambient snowpacks.

Thank you very much for this comment. We are indeed only considering absorption and not scattering by snow-borne impurities. As stated in response to reviewer #3, we are also focusing on "effective black carbon", and are thus focused on just the absorption component of "extinction". We will re-word the revised manuscript appropriately. On a side note, for studies using the LAHM measurement technique, we do have some filters collected in Nepal (by co-author UH) that have very high quantities of scattering particles that have extremely little absorption.

6) The suggested sampling strategy is to collect one sample that isolates as best as possible the surface layer of particles, while collecting as little of the surface snow as possible, then collecting a single sub-surface sample (depth not specified). However, vertical variations in the MMR of LAPs in the sub-surface snow may also be important. For example, a snowpack that has significant accumulation of particles on the surface might also have a thin melt- or wind-crust that has quite a different LAP MMR than the snow immediately below that. Or perhaps there was a snowfall of a few cm, then a long period of no new snow but significant dry deposition, then another cm of relatively clean new snow – then the accumulation of particles on that snow surface. While vertical variations in the MMR of LAP in the snow will be most important near the snow surface, all such vertical variations will matter down to the penetration depth of sunlight (approx. 10cm, +/- depending on LAP concentrations and snow grain size). The proposed sampling strategy is certainly a good *minimum* requirement for snow with very high concentration of LAPs on top of a snowpack. However, a single sub- surface sample as a recommended strategy is only appropriate where the snow below these particles has very uniform LAP MMR – a very specific configuration that will occur in only a very limited number of cases. If an improved sampling strategy is to be proposed, it would be better to account for a larger range of possible cases by specifying that, whenever possible to do so, multiple sub-surface samples should be collected. If visible layers are present in the snow, sample depths could be determined visually; where no visible vertical variations are present, if several sub-surface layers can be collected it would be best to sample thinner layers towards the snow surface (e.g. collecting samples of 2cm depth, 3cm deep, then 5cm deep for the case where three sub-surface layers could be collected, to cover the full 10cm that light is likely to penetrate).

This is very true and is what we address in the conclusions around line 270 (not to the extent that you do, but we will expand the text). In Colorado, where the lead author has done a lot of sampling, more often than not, there is a layer of fresh snow over the top of a "former" surface that has an LAP accumulation layer. Especially in a situation with 2-3 cm of fresh snow over a very dirty surface, it is important to characterize the snow at all layers. An important point that will be elucidated in the updated manuscript will be that layers need to be treated as layers (no matter where they are in the snowpack). If a 1 mm thick layer is really dirty no matter where it is in the snowpack, it needs to be treated as a 1 mm thick layer with an appropriate LAP mass in the layer and it should not be averaged over a depth substantially more than it occupies. There is no "one size fits all" sampling strategy. The strategies suggested in the manuscript are meant to be "one size fits most". The authors all work with citizen science groups, so a one size fits most strategy is most likely to get decent results with untrained data collectors. On line 270, the manuscript states that "much more complicated snowpacks" need more complicated treatments and our hope is that scientists who need more accuracy will do more detailed sampling and with the knowledge gained from the manuscript, collect snow with different layers being treated as layers with substantial depth or with minimal depths (more 2 dimensional). We will emphasize the importance of sampling layers appropriately in the updated manuscript.

7) A simple thought experiment is sufficient to know it must be the case that concentrating LAPs at the surface of the snowpack will have a greater effect on albedo than will mixing the same amount of LAPs throughout some depth of the snowpack. This does not need to be qualitatively demonstrated via experiment. What *is* needed is quantitative verification of the proposed theoretical method of calculating albedo in the case where LAPs are on the snow surface, using the sampling strategy and calculation method proposed. The experiment described in Section 3 (results shown in Figure 3) doesn't provide such a quantitative test. Thus, I question the value of including discussion of this experiment in the paper.

As stated earlier, we feel that the qualitative demonstration of the results (the experiments shown in figure 3) are important to show, but due to the constraints, it was not possible to conduct the experiments in a way that could be very accurately quantified. Thus, we would consider moving this section to being supplemental to the publication and not part of the publication.

The revised manuscript will show that the SNICAR values trend towards (but not exactly to) the equation calculated results as the surface layer in SNICAR trends thinner and thinner for the Vallunaraju case. Unfortunately, we don't have albedo measurements for that day (it was a citizen science expedition) but the albedo was definitely closer to 0.3 than 0.7 (the SNICAR calculations for a 10 micron layer and a 1 cm layer respectively).

8) If the experimental results are going to be included two things need to be addressed:

a) The setup had the FieldSpec located 30cm above the snowpack, viewing an area 20-30cm in diameter to which LAPs had been added. I am assuming the FieldSpec has something like a cosine-weighted input. (Such details should be specified). The instrument therefore would have been viewing an area that includes snow without added LAP, correct? This is not mentioned or accounted for.

b) The "clean snow" (snow to which LAP has not been added) is reported as having an eBC MMR of 12 ng/g. In the four cases shown in Figure 3, the ~500nm albedo of this snow varies from <0.8 to >0.95. An MMR of 12 ng BC/g snow isn't sufficient to explain a 500nm snow albedo of <0.8. In addition there's either actual variations in the snowpack or significant measurement uncertainties that are not being accounted for here, demonstrating that this is a very qualitative analysis.

Agreed, as discussed, qualitative, but we feel it is worth including given that it demonstrates the effect, although qualitatively only.

---

## Author Comment (AC3) · 13 Jan 2020

The authors would like to thank the three anonymous reviewers for their comments and suggestions. In the revised manuscript, we will address all of these comments. Detailed responses to how these comments will be addressed are included in this document. In reading the reviews, there were a few topics that we feel require fundamental (if minor) changes to the manuscript to increase understanding. We will address those here and refer to these comments later when responding to individual comments.

The terminology of a "layer" in the manuscript will be changed in the revised manuscript. We regularly referred to "surface layers" of LAPs in the manuscript, but it seems easy to confuse this with different "layers" in the snow. The difference being that "layers" in the snow are generally assumed to have macroscopic thicknesses on the order of centimeters while a "surface layer" of LAPs in the manuscript refers to a layer of non-ice particles, that are not mixed into a macroscopic snow layer. In the revised manuscript the LAP layer will be referred to as a "coating" or "surface coating" of LAPs to distinguish it from snow layers including layers of snow with LAPs mixed in. We feel that this change will make the discussion and interpretation more clear.

Below is a new table, a version of which will go in the revised manuscript. The Vallunaraju case is used to demonstrate the impact of different layer thicknesses. SNICAR was run with the same total eBC with the eBC being distributed evenly through different layer thicknesses (MMR increased as layer thickness decreased conserving total eBC). The visible albedo calculated by SNICAR (assuming a total snowpack depth of 10 meters) shows large differences depending on how one partitions the LAPs. Note that the modeled albedo using equation 1 in the manuscript is 0.38. Also, note that this was a much more extreme case than that shown in Figure 1, thus, although we don't have direct albedo measurements from the case, we suspect that the actual albedo was closer to 0.3 than 0.7. This table will be included in the manuscript as well as discussion of these results.

| layer thickness (m) | LAP conc (ng/g) | LAP below | visible albedo |
| --- | --- | --- | --- |
| 0.1 | 800 | 0 | 0.8914 |
| 0.01 | 8000 | 0 | 0.7013 |
| 0.001 | 80000 | 0 | 0.4189 |
| 0.0001 | 800000 | 0 | 0.3116 |
| 0.00001 | 8000000 | 0 | 0.2974 |
| 0.000001 | 80000000 | 0 | 0.2983 |

All three reviewers mention the use of the asymmetry parameter is erroneous and incorrectly defined. In the revised manuscript, the surface reflection will be redone more realistically. The 15% estimate in the original manuscript is likely too high based on initial calculations suggesting that this factor is less important.

**Anonymous Referee #3**

The manuscript makes the point that differentiating layers of light absorbing particle (LAP) mass mixing ratios (MMR), particularly the surface layer, is necessary to derive a realistic value for the resulting surface albedo. A theoretical justification is provided which is underlined by a qualitative field experiment. Based on this, a sampling protocol is suggested and instructive online material is provided.

The manuscript raises the important point that bulk LAP MMR in snow samples can underestimate the MMR in the surface layer where LAPs can accumulate due to melting snow, sublimation or dry deposition of LAPs. The surface layer is obviously the most important layer for the resulting surface albedo. It is hence important to characterize the surface layer separately and harmonize the sampling protocol in the community to produce more comparable results. While this issue certainly deserves dissemination and is a topic for 'The Cryosphere', the manuscript does not present any new results per se, but is rather method focused. This makes it perhaps better suited for a 'Technical Note', rather than a full scientific manuscript, especially because the presented field experiment results are very qualitative. In addition, several points will have to be addressed to correct and complete the work, hence I suggest major revisions.

Our original intention was to submit it as a technical note, but over time the manuscript grew to the size of an article. As you mention (and as is discussed in replies to Reviewers 1 and 2, the field experiment is very qualitative and we would consider moving it to be supplementary materials. Though qualitative, they do demonstrate the impact consistently.

General comments:

1) The manuscript will greatly benefit from more context: Why is the accurate determination of snow surface albedo important? In which models and how are such measurements used? What is the use of a point and snapshot measurement as described in this work? A common challenge is the upscaling spatially and temporally with such field measurements. Spatial heterogeneity of LAP MMR in surface snow is one issue. The other aspect is, which has also not been discussed, that LAP do not only accumulate at the surface under certain conditions, but that also new snowfall can reduce the MMR of LAP at the surface temporarily. So the temporal dimension is also very important to consider as high and low LAP MMR conditions at the surface might alternate. To be able to model this a number of processes need to be characterized towards their importance for surface albedo, and it will be important to convey where in this more complex consideration the value of this work lies. There is a vast amount of literature and some of it should be reflected in the introduction.

The original purpose of the research was to develop a model for the evolution of snow albedo over time. It quickly became apparent that quasi-2D layers were extremely important. The LAP concentration on the surface of a snow pack in the future is a function of the current LAP concentration and the quantity of snow that melts and any dry deposition. This then drives the next time period's melting. So, in a sense, in addition to being a point measurement, this could describe input to a model.

The "clean snow on top of a dirty layer" scenario is discussed in the sampling strategies section in the conclusions (line 267). This is common in many places where members of the author team have sampled (Colorado and Peru). Suggested sampling is to collect the fresh snow to determine the MMR, then clean off the fresh snow and treat the surface as having an LAP coating, then sample the sub surface snow below that layer.

2) This work is entitled "The measurement and impact of light absorbing particles on snow surface", however it focuses on the 'measurements' rather than the 'impact' (see need for additional context above). In addition, the work focuses on black carbon (BC) rather than the diverse types or mix of LAP to which mineral dust, brown carbon, organic carbon, microorganisms and others belong as well. Measuring all of these will further complicate albedo determination and is beyond the scope of this manuscript. But the mix is an important feature of real snow samples and will vary in composition by location and season. Hence, either a discussion of how this could be addressed and how sensitive albedo calculations are to it

is needed in the manuscript, or at the very least pointing out the issue is needed, and the title should be changed to "The measurement of black carbon in snow surfaces".

This comment brings up a lot of interesting details and problems. If we are interested in the light absorption capability (and thus it's impact on albedo) of anything, it is necessary to define appropriate units. Unfortunately, historically, the light absorbing particle in snow community has settled on something completely unrelated to light absorption: a mass unit of black carbon. As you state in your comment #6, this is not a constant. We should be discussing the impact (absorption cross section of particles) as opposed to a mass of something that has variable absorption efficiencies. Note that the manuscript uses effective black carbon (eBC), not black carbon throughout and we define it as in Grenfell et al (2011): the mass of black carbon that absorbs the same amount of light as the present light absorbing particles. And specifically, we relate measurements to Fullerene soot type black carbon since Fullerene soot is well characterized by the scientific community and is commonly used for calibration. As pointed out by reviewer 2, we do not address the light scattering capabilities of the particles.

The revised manuscript will include a lot more information on the impact of different distributions of LAPs at or near the surface. With the caveat that we are not discussing the impact of light scattering by impurities in snow, we feel that "Light absorbing particles" is definitely more appropriate than "black carbon" in the title.

3) Some more precise definition of the "2D surface layer" is needed. Since it practically cannot be 2D more details on the depth in this particular study need to be included and it also should be discussed how the depth of the surface layer can vary by location. There might be examples where the LAP enrichment happened within millimeters, while for other locations a centimeter depth would describe the surface layer equally well. Also, there should be more information given on the depth to which light penetrates the snow and on which parameters this depends (snow characteristics, LAP MMR etc.), to provide a reasonable range of the overall sampling depth that should be covered in different layers. One assumption this work makes is that under the surface layer LAPs are mixed homogeneously, which most likely is a special case. In most instances, more layers will need to be defined. This is briefly mentioned at the end of the manuscript.

Our assumption is that dry deposition does not become mixed into a thick layer of snow. Obviously, it can potentially trickle down between crystals, but this isn't likely to take it too far. When snow melts, studies have shown that some impurities (20%) can wash out of a snowpack, but obviously (look at snow in the late spring) a lot of impurities stay at the top. As noted in the general replies, the revised manuscript will include a lot more information and calculations on the thickness of the surface layer.

4) The definition of the asymmetry parameter 'g' is not correct, please revise.

As stated in the initial comments, the surface reflection will be redone and will likely come out lower thus reducing the impact of this factor.

5) The albedo calculation with SNICAR requires information on snow grain size, a critical parameter for the snow albedo (in each of the layers). Not only the spread and MMR of LAP at the surface are important, but also the changing snow grain size during the LAP accumulation process. There is no information in the manuscript about the measurements or assumptions made. Please add this. The same is true for the solar zenith angle.

This information will be added to the revised manuscript. As stated in response to other reviewers, the emphasis of this manuscript is understanding the impact of the surface coating or very thin opaque layers

in a snowpack. The impact of the snow properties is a different topic so we don't go into it much as we don't want to take away from the message of the manuscript.

6) The optical properties of BC depend on its size. There is literature (e.g. Schwartz et al., 2013, DOI: 10.1038/srep01356) showing that BC size in snow is larger than atmospheric BC due to (post-)deposition processes. This shifts the MAC to smaller values. This process is particularly relevant for situations where BC accumulates at the snow surface due to melting or sublimation, or when the MMR is so high that particles stick to each other. This work does not take the size of BC into account and also does not discuss how the artificial soot doping of the snow in the section 3 experiments compares to atmospheric soot that is deposited. While it is evident that this information cannot be retrieved anymore and is also not trivial to derive, at least a discussion needs to be included because the effect partly compensate the estimated albedo reduction reported here.

Yes, we agree. This is why we included calculations using a MAC of 5 rather than the standard Fullerene soot MAC of 9 (the current table 1). As stated in response to one of your earlier comments, the manuscript uses eBC, effective black carbon, so the particles absorb light as efficiently as "X" amount of Fullerene soot (with an assumed MAC of 9). Note that LAHM measures eBC, not BC, so the measurements related to Figure 3 are not related to the actual mass. (Activated charcoal likely has a much larger size than atmospheric BC).

7) The paper would benefit strongly from a more comprehensive uncertainty analysis that includes for example variation of the following parameters: snow grain size, MAC value, estimate of the LAP covered area (because it is not clear based on which consideration the overlap is estimated in the manuscript, should be made clear), the surface layer thickness etc. This will be very instructive to learn where the largest uncertainties come from, and for the sampling community for where to pay attention for most accurate measurements. The would particularly benefit section 3, which is qualitative only, and hence does not add new information to the manuscript per se. The information section 3 provides is also provided in section 2, the theoretical consideration.

More uncertainty analysis will be added to the revised manuscript.

8) Figure 2 should be made more accurate. The variables mentioned in the text should appear, the transmitted and reflected radiation should be marked by arrows. Snow grains and layers are important features as well as an indication of the light penetrating depth.

Figure 2 will be remade as suggested.

9) The suggested sampling strategy is not new per se, it is rather a refinement of an existing method.

True, in the revised manuscript we will re-word this.